# PROVABLE GUARANTEES FOR FLOW-BASED GENERATIVE MODELS IN TIME SERIES

## ABSTRACT

Recent studies suggest utilizing generative models instead of traditional auto-regressive algorithms for time series forecasting (TSF) tasks. These non-auto-regressive approaches involving different generative methods, including GAN, Diffusion, and Flow Matching for time series, have empirically demonstrated high-quality generation capability and accuracy. However, we still lack an appropriate understanding of how it processes approximation and generalization. This paper presents the first theoretical framework from the perspective of flow-based generative models to relieve the knowledge of limitations. In particular, we provide our insights with strict guarantees from three perspectives: **Approximation**, **Generalization** and **Efficiency**. In detail, our analysis achieves the contributions as follows:

- By assuming a general data model, the fitting of the flow-based generative models is confirmed to converge to arbitrary error under the universal approximation of Diffusion Transformer (DiT).
- Introducing a polynomial-based regularization for flow matching, the generalization error thus be bounded since the generalization of polynomial approximation.
- The sampling for generation is considered as an optimization process, we demonstrate its fast convergence with updating standard first-order gradient descent of some objective.

## 1 INTRODUCTION

Generative models have revolutionized machine learning by enabling the creation of highly realistic and diverse content across various domains. In particular, diffusion-based approaches (Ho et al., 2020), Generative Adversarial Networks (Karras et al., 2021), and flow matching methods (Lipman et al., 2023) have emerged as powerful tools for data synthesis and augmentation. These methods leverage sophisticated architectures to learn complex probability distributions and transform random noise into structured, meaningful outputs. For example, text-to-image models translate textual descriptions into compelling visual artworks or photographs (Zhang et al., 2023), while recent advances in text-to-video frameworks produce coherent and temporally consistent video content (Ho et al., 2022). Discrete flow matching (Gat et al., 2024) extends continuous-time flow-based modeling to discrete settings by carefully aligning discrete probability distributions via flexible transformations, thereby broadening the applicability of flow-based generative models to high-dimensional discrete domains such as language and code. As these techniques continue to evolve, the ability of generative models to capture intricate data structures and produce high-quality samples underscores their broadening influence in artificial intelligence research.

Among all these data types, time series data, found in fields like finance, healthcare, and climate science, constitutes a critical yet challenging domain for forecasting and analysis (Box & Jenkins, 1976). Given its temporal dependency and noisy nature (Box et al., 2015), time series poses unique obstacles that often exceed the complexities encountered in static data settings. By establishing the NP-hardness of computing a mean in dynamic time-warping spaces, (Bulteau et al., 2020) highlights key computational challenges in time series analysis. Nonetheless, the powerful capabilities of generative models have proven effective in tackling these challenges, offering promising solutions on time series data. By learning the underlying distribution of time series trajectories, generative

approaches can capture both signal and noise components, thereby producing more robust forecasts and generalizations. Indeed, the recent success of GAN (Jeon et al., 2022), diffusion (Rasul et al., 2021a; Tashiro et al., 2021), and flow-based models (Zhang et al., 2024b) in time series highlights their growing appeal, as these tools exhibit strong empirical performance across diverse application scenarios (Li et al., 2022; Wang & Ventre, 2024; Tian et al., 2024). Consequently, the burgeoning research on generative models for temporal data generation and forecasting stands at the forefront of machine learning, offering transformative potential for both academia and industry.

Although such generative models show remarkable performance when applied to time series, our theoretical understanding of their success remains limited. Researchers have begun questioning what fundamental principles govern their approximation capabilities and how well they generalize under real-world data conditions (Zhang et al., 2024a; Fukumizu et al., 2025). Without a solid theoretical framework, it is difficult to fully trust and optimize these methods, and their reliability in safety-critical domains becomes a concern. While empirical evidence consistently demonstrates their potential, the absence of a rigorous conceptual foundation obscures deeper insights into model selection, hyperparameter tuning, and design strategies. Indeed, bridging this gap between practical efficacy and theoretical clarity is an urgent priority, which motivates our efforts to explore flow-based generative models for time series and provide meaningful error bounds and generalization guarantees.

In this work, we propose a strict framework to analyze the generative models for time series generation, especially the flow-based generative models (Hu et al., 2024; Yuan & Qiao, 2024). It involves three parts:

- *Approximation.* Theorem 5.4 confirm that flow-based generative models converge to arbitrary approximation error under the universal approximation capability of DiT in Section 5.
- *Generalization.* Theorem 6.2 derive bounded generalization error guarantees, leveraging the inherent approximation properties of orthogonal polynomial bases to ensure robustness against noise and distribution shifts in Section 6.
- *Efficiency.* Theorem 7.7 in Section 7 establishes fast convergence guarantees through gradient descent dynamics, demonstrating that our framework achieves efficient generation while maintaining theoretical stability.

**Roadmap.** In Section 2, we review relevant related work. Section 3 introduces key background concepts and the problem setup. In Section 4, we present the framework for time series generation using flow matching. Section 5 discusses the approximation results, while Section 6 covers generalization results. Section 7 examines efficiency results. We discuss limitation in Section 8. Finally, we conclude our paper in Section 9.

## 2 RELATED WORK

**Generative Models.** Generative models have emerged as a powerful framework for learning complex data distributions, encompassing methods such as Variational Autoencoders (VAEs) (Kingma & Welling, 2014; Rezende et al., 2014), Generative Adversarial Networks (GANs) (Goodfellow et al., 2014; Arjovsky et al., 2017; Gulrajani et al., 2017; Karras et al., 2021), and diffusion-based approaches (Sohl-Dickstein et al., 2015) that iteratively refine noisy samples. VAEs introduce a latent-variable formulation with an encoder-decoder architecture to learn a smooth latent space, while GANs employ a minimax game between generator and discriminator to capture sharp data distributions. Recent diffusion approaches, such as Denoising Diffusion Probabilistic Models (DDPM) (Ho et al., 2020), progressively destroy data by adding noise and then reverse the process via learned denoising steps. Score-based methods (Song & Ermon, 2019; Song et al., 2020b) generalize this process by estimating the gradient (score) of the data density to generate samples through stochastic differential equations. Normalizing flows (Rezende & Mohamed, 2015; Papamakarios et al., 2021) take an alternative route by constructing invertible transformations with tractable Jacobians, enabling exact likelihood computation. More recently, novel paradigms such as flow matching (Lipman et al., 2023) and rectified flow (Liu et al., 2023) have emerged, aiming to simplify sampling via direct trajectory-based transformations. In parallel, advancements in Diffusion Probabilistic Model (DPM) solvers (Lu et al., 2022a;b) further optimize the sampling process, reducing computational

overhead while preserving generative fidelity. Collectively, these developments highlight a vibrant research landscape, where systematic improvements and new theoretical insights continue to push the boundaries of generative modeling (Chen et al., 2025a;b; Gong et al., 2025b; Cao et al., 2025a; Gong et al., 2025a; Cao et al., 2025b; Liang et al., 2025).

**Generative Models for Time Series Forecasting.** Concurrently with advancements in generative models, a powerful and often more computationally efficient paradigm has emerged for the generative modeling of time series, with notable examples including Generative Adversarial Networks (GAN) (Yoon et al., 2019), Variational Autoencoders (VAE) (Desai et al., 2021), and normalizing flows (Rasul et al., 2021b). More recent works demonstrate that flow matching is a powerful technique for time series modeling. For example, (Tamir et al., 2024) incorporates conditional Gaussian processes as informed prior distributions and achieves state-of-the-art probabilistic forecasting results across eight real-world datasets. To enhance scalability and stability, (Zhang et al., 2024b) introduces a simulation-free training algorithm for Neural Stochastic Differential Equations. (Kollovieh et al., 2025) simplifies the generation process based on optimal transport principles and couplings. Based on rectified flow, (Hu et al., 2024) avoids iterative sampling and complex noise schedules often found in diffusion models. Within the framework of flow matching (Lipman et al., 2023; Gao et al., 2025), diffusion modeling has also been successfully applied to time series. For example, to leverage the unique properties of time series data, (Shen et al., 2024) employs seasonal-trend decomposition to extract fine-to-coarse trends from the time series for forward diffusion. While these works demonstrate remarkable empirical advancements in diffusion and flow-based models for time series forecasting, consistently outperforming prior methods, they also highlight a critical gap in theoretical understanding regarding their underlying mechanisms, generalization, and convergence. This deficiency thereby amplifies the need for further research to ensure reliable deployment and guide future principled development.

## 3 PRELIMINARY

This section introduces the theoretical background we aim to address in this paper. In detail, we introduce the key notations and definitions for window sizes, pseudoinverses, and other fundamental concepts in Section 3.1. In Section 3.2, we formally define the time series forecasting and imputation problem by presenting the data model, assumptions on smooth signals and Gaussian noise, and the objective function.

### 3.1 NOTATIONS

We use $[n]$ to denote the set $\{1, 2, \cdots, n\}$. We use $\mathbb{E}[]$ to denote the expectation. We use $\|A\|_F$ to denote the Frobenius norm of a matrix $A \in \mathbb{R}^{n \times d}$, i.e. $\|A\|_F^2 := \sum_{i \in [n]} \sum_{j \in [d]} |A_{i,j}|^2$. We use $\| \cdot \|_p$ to denote the $\ell_p$ norm of a vector $x \in \mathbb{R}^d$, i.e. $\|x\|_p^2 := (\sum_{i \in [d]} |x_i|^p)^{1/p}$. We use $\| \cdot \|_\infty$ to denote the $\ell_\infty$ norm of a vector $x \in \mathbb{R}^d$, i.e. $\|x\|_\infty := \max_{i \in [d]} |x_i|$. We use a positive integer $N_x$ to denote the window size of input data, and a positive integer $N_y$ to denote the window size of output data. Especially, we have $N_x \gg N_y$ and denote $N := N_x + N_y$. The function $\lambda_{\min} : \mathbb{R}^{d_1 \times d_2} \to \mathbb{R}$ takes any matrix $A \in \mathbb{R}^{d_1 \times d_2}$ as input and outputs the smallest singular value of matrix $A$. We use $|\cdot|$ to represent the size of a set. We use $e_\tau \in \mathbb{R}^N$ to denote the $N$-dimensional one-hot vector with the $\tau$-th entry is 1 for any $\tau \in [N]$. For any matrix $A \in \mathbb{R}^{d_1 \times d_2}$, we use $A^\dagger \in \mathbb{R}^{d_2 \times d_1}$ to stands for its pseudoinverse. We say a matrix $A$ is positive definite (PD) once its smallest singular value is positive, $\lambda_{\min}(A) > 0$.

### 3.2 PROBLEM DEFINITION

**Data Distribution and Training Set.** We first define the data model of time series:

**Definition 3.1** (Data model). *We consider a continuous target function in Sobolev-RKHS (Reproducing kernel Hilbert space) of arbitrary smoothness $s \geq 1.5$ that we aim to learn, denoted $f^* \in H^s(\mathcal{X})$, where $\mathcal{X} = [0, T_{\max}]$ for considerable time $T_{\max} \gg 0$ and we note that $\|f^*\|_{H^s(\mathcal{X})} \leq \Theta(T_{\max})$. The data distribution is given by:*

$$\mathcal{D} = \{(t, f^*(t) + \xi), t \in \mathcal{X}, \xi \sim \Xi\},$$

*where $\Xi$ is some noise distribution that is centered at zero and the variance is $v \geq 0$. Notably, the following properties hold:*

- *Property 1. $f^*$ is Lipchitz smooth, we denote the smoothness as $L_0$ (Part 1 of Lemma A.1).*

- *Property 2. Denote the failure probability $\delta \in (0, 0.1)$, then with a probability at least $1 - \delta$, we have $|f^*(t) + \xi| \leq \sqrt{\frac{v}{\delta}}$ (Part 2 of Lemma A.1).*

**Remark 3.2.** *We emphasize that the setting of Definition 3.1 is mild and widely applicable in the machine learning fields, especially in time series, as it can perfectly fit all data distributions mixing with noise and target functions (Kitagawa, 1987; Ozaki & Iino, 2001; Middleton, 2002; Cryer & Chan, 2008).*

Therefore, we state the definition of the data sampling method as follows:

**Definition 3.3.** *For any time duration $[T_{\text{left}}, T_{\text{right}}]$ that uniformly chosen from $[0, T_{\max}]$, we define the grid points as:*

$$\mathcal{G}(T_{\text{left}}, T_{\text{right}}) = \{T_{\text{left}}, T_{\text{left}} + \frac{T_{\text{right}} - T_{\text{left}}}{N - 1}, \cdots, T_{\text{right}}\}, |\mathcal{G}| = N.$$

*We denote the dataset size as $m$, then the training set is:*

$$\mathbb{D} = \left\{ [t_j, f(t_j) + \xi_j]_{j=1}^N \in \mathbb{R}^{2 \times N} \middle| t_j \in \mathcal{G}(T_{\text{left}}, T_{\text{right}}), \xi_j \sim \Xi, T_{\text{left}}, T_{\text{right}} \sim \mathcal{U}[0, T_{\max}] \right\}_{i \in [m]}.$$

**Review on Classical Time Series Modeling.** Obtaining $\mathbb{D}$ in Definition 3.3, the classical methods to model time series usually split $[t, f] \in \mathbb{D}$ to $f_x \in \mathbb{R}^{N_x}$ as model input and $f_y$ ideal output with some certain strategy, we first define two observation matrices as follows:

**Definition 3.4.** *We define the indices set $\mathcal{I} = [N]$, where input indices set is its certain subset $\mathcal{I}_x \subset \mathcal{I}$ and $|\mathcal{I}_x| = N_x$, and the target indices set $\mathcal{I}_y = \mathcal{I}/\mathcal{I}_x$, $|\mathcal{I}_y| = N_y$. We use $\mathcal{I}_k$ to denote the $k$-th element of $\mathcal{I}$ for $k \in [N]$. For any data $[t, f] \in \mathbb{D}$, we define the input observation matrix $M_x \in \mathbb{R}^{N_x \times N}$, where its $i$-th row $M_{x,i} = I_{N, \mathcal{I}_{x,i}}$, likewise, $M_{y,i} = I_{N, \mathcal{I}_{y,i}}$.*

Then we give the definition of the regression problem:

**Definition 3.5.** *Let observation matrices $M_x \in \mathbb{R}^{N_x \times N}$ and $M_y \in \mathbb{R}^{N_y \times N}$ be defined as Definition 3.4. For each data point $[t, f] \in \mathbb{D}$, we let $f_x := M_x \cdot f$ and $f_y := M_y \cdot f$. Given a model class set $\mathcal{F} \subset H^s(\mathcal{X})$, the regression training objective of classical time series modeling, e.g. AR methods, is defined as mean square error, such that:*

$$L_{\text{classical}}(F) := \underset{(t,f) \in \mathbb{D}}{\mathbb{E}}[\|F(M_x \cdot f, M_x \cdot t) - M_y \cdot f_y\|_2^2], F \in \mathcal{F}.$$

*Therefore, the expected risk of $F$ is ($\Delta > 0$ is some fixed sample size)[1]:*

$$\mathcal{R}(F) := \underset{t \in [0, T_{\max} - N \cdot \Delta], \xi \sim \Xi}{\mathbb{E}}[\|F(f_{x,t}, \tau) - f_{y,t}\|_2^2],$$

*where $f_x(t) = M_x f_t, f_y = M_y f_t, f_t = [f^*(t + (j-1) \cdot \Delta) + \xi_j]_{j=1}^N, \tau = [t_j + (j-1) \cdot \Delta]_{j=1}^N$.*

The error decomposition is trivially given by :

$$\mathcal{R}(F) = \Delta \mathcal{R}(F) + \mathcal{R}(F^*)$$
$$= \underset{t \in [0, T_{\max} - N \cdot \Delta], \xi \sim \Xi}{\mathbb{E}}[\|F(f_{x,t}, \tau) - [f^*(t + (j-1) \cdot \Delta)]_{j=1}^N\|_2^2] + v.$$

## 4 FLOW MATCHING FOR TIME SERIES GENERATION

In this section, we introduce the core framework and methodology for time series generation using conditional flow with polynomial regularity, followed by the training objective and a sampling algorithm. We first introduce the polynomial approximation that our conditional flow relies on in Section 4.1. Here we explore polynomial approximation bases, highlighting their orthogonality,

---

[1]Usually, we fix $T_{\text{right}} - T_{\text{left}}$, then $\Delta = \frac{T_{\text{right}} - T_{\text{left}}}{N-1}$.

positive definiteness, and strong approximation capabilities in modeling time series data. In Section 4.2, we define the conditional flow for time series generation, introducing the time-dependent mean and standard deviation functions, and the polynomial regularization of the flow. In Section 4.3, we specify the training objective based on the Flow Matching framework, defining the loss function and providing the closed-form solution for the optimal model.

## 4.1 POLYNOMIAL APPROXIMATION

In the following, we discuss a group of specific polynomial bases. Since the strong approximating ability to differentiable functions like the Fourier approximation (usually converging to an arbitrary error with a sufficiently high order), previous works (Yuan & Qiao, 2024; Hu et al., 2024) apply such an approach as a regularization method that provides the model with prior knowledge. In the range of this paper, we define a sequence of specific orthogonal polynomial bases as

**Definition 4.1** (Orthogonal polynomial bases). *Let $n$ be the number of orders of the polynomials. We define the orthogonal polynomial bases $P$ as $P := [P_1, P_2, \cdots, P_n] \in \mathbb{R}^{N \times n}$ where each column $P_i \in R^N$ for any $i \in [n]$ is a polynomial basis. It satisfies that (1) The degree of $P_i \in \mathbb{R}^N$ for any $i \in [n]$, denotes $\deg(P_i) = i - 1$. (2) Each polynomial basis is orthogonal due to some measurement $\ell$. Formally, $\langle P_i, P_j \rangle_\ell = 0$. (3) $P$ is positive definite (PD), such that $\lambda := \lambda_{\min}(P) > 0$. (4) The upper bound on $\ell_\infty$ norm of $P$ is $\exp(O(nN))$.*

The approximating capability of polynomial approximation is obvious. To show that, we first introduce a tool from previous work:

**Lemma 4.2** (Proposition 6 in (Gu et al., 2020)). *If the following conditions hold: Let $f : \mathbb{R}^{\geq 0} \to \mathbb{R}$ be a differentiable function. Let $g_t := \mathrm{proj}_t(f)$ be its projection at time $t$ with maximum polynomial degree $N - 1$. Assume $f$ is $L$-Lipschitz. Then we have $\|f - g_t\|_2 = O(tL/\sqrt{N})$.*

Apply the above lemma, we can show

**Lemma 4.3.** *Let $g : \mathbb{R}^{\geq 0} \to \mathbb{R}$ be the signal. Let $f' := [g(\tau \cdot \Delta)]_{\tau=1}^N$ be the sample for some signal $g$, where $\Delta$ is the sample step size. Then we have $\|PP^\dagger f' - f'\|_2 = O(NL_0/\sqrt{n})$.*

*Proof.* This result follows from Lemma 4.2. □

## 4.2 CONDITIONAL FLOW WITH POLYNOMIAL REGULARITY

Now we introduce the design of the conditional flow in this paper, which relies on the polynomial bases we defined in the previous subsection. We first introduce a linear projection matrix as follows:

**Definition 4.4.** *Let the polynomial basis $P$ be defined in Definition 4.1, and we denote the observation matrix $M_y \in \mathbb{R}^{N_x \times N}$ as Definition 3.4. We define the matrix $G \in \mathbb{R}^{N_y \times n}$ as $G := M_y P$.*

Specifically, we define the time-dependent mean of a Gaussian distribution satisfying an ordinary differential equation. It is also called our polynomial regularization.

**Definition 4.5** (Time-dependent mean of Gaussian distribution). *Let $f = [f_x^\top, f_y^\top]^\top \in \mathbb{R}^N$ be defined as Definition 3.1. Let $\alpha \in (0, 1)$ be some constant. Let $G$ be defined in Definition 4.4. We define the time-dependent mean of Gaussian distribution as $\mu : [0, T] \times \mathbb{R}^N \to \mathbb{R}^{N_y}$, which satisfied the ODE that $\mu_t'(f) = \alpha \cdot GG^\top(GG^\dagger \psi_t(f) - f_y)$,*

Meanwhile, we define the time-dependent standard deviation as controlling the uncertainty in the distribution, starting from a broad variance and gradually narrowing to a minimum value, which helps regulate the learning dynamics and stabilize the model.

**Definition 4.6** (Time-dependent standard deviation). *Let $f = [f_x^\top, f_y^\top]^\top \in \mathbb{R}^N$ be defined as Definition 3.5. Let $t \sim \mathrm{Uniform}[0, T]$. Let $\sigma_t : \mathbb{R}^N \to \mathbb{R}$. We define the minimum standard deviation $\sigma_{\min}$ as $\sigma_{\min} := \sigma_1(f)$. We define the time-dependent standard deviation $\sigma$ as $\sigma_t(f) := 1 - (1 - \sigma_{\min})t$.*

The flow matching for time series generation (Galib et al., 2024) defines a flow $\psi : [0, 1] \times \mathbb{R}^N$ taking time $t$ and time series data as input, matching $\psi_0(f) \sim \mathcal{N}(0, I_{N_y})$ at the beginning and $\psi_1(f) = f_y$ in the end, and then applying some neural networks to fit this distribution-to-distribution process. The detailed definition is given by:

**Definition 4.7.** *Let $f = [f_x^\top, f_y^\top]^\top \in \mathbb{R}^N$ be defined as Definition 3.5. Let $\mu_t(f)$ be defined in Definition 4.5. Let $\sigma_t(f)$ be defined in Definition 4.6. Let $z \sim \mathcal{N}(0, I_{N_y})$ be the sample. We define the flow $\psi_t(f) \in \mathbb{R}^{N_y}$ as follows: $\psi_t(f) := \sigma_t(f) \cdot z + \mu_t(f)$.*

### 4.3 Training Objective with Polynomial Regularity

We slightly deviate from standard notation by defining the model function $F_\theta : \mathbb{R}^{N_y} \times \mathbb{R}^{N_x} \times [0, 1] \to \mathbb{R}^{N_y}$, parameterized by $\theta$, to capture the polynomial regularized conditional flow $\psi_t(f)$ introduced in Definition 4.7. This function takes the flow along with a temporal input to infer the corresponding vector field. The training procedure employs the Flow Matching framework (Lipman et al., 2023), which strives to shrink the discrepancy between the model's estimates and the actual derivative of the flow.

Consequently, we define the training objective as the expected squared $\ell_2$ norm of the discrepancy:

**Definition 4.8** (Training Objective). *Let $t \sim \mathsf{Uniform}[0, T]$. Let $f = [f_x^\top, f_y^\top]^\top \in \mathbb{R}^N$ be defined as Definition 3.5. Let $z \sim \mathcal{N}(0, I_{N_y})$ be the sample. Let $\psi_t(f)$ be defined in Definition 4.7. Let $F_\theta : \mathbb{R}^{N_y} \times \mathbb{R}^{N_x} \times [0, T] \to \mathbb{R}^{N_y}$ be the model with parameter $\theta$. We define the training objective as*

$$\mathcal{L}(\theta) := \mathop{\mathbb{E}}_{z,t,f}[\|F_\theta(\psi_t(f), f_x, t) - \frac{\mathrm{d}}{\mathrm{d}t}\psi_t(f)\|_2^2].$$

We then provide the closed-form solution for $F_\theta$ that achieves the minimum of $\mathcal{L}(\theta)$ as follows:

**Theorem 4.9** (Informal version of Theorem B.1). *Let $\mathcal{L}(\theta)$ be defined in Definition 4.8. Let $z \sim \mathcal{N}(0, I_{N_y})$. Let $t \sim \mathsf{Uniform}[0, T]$. Let $f_x$, $f_y$ be defined in Definition 3.5. Let $G$ be defined in Definition 4.4. Let $\sigma_{\min}$ be defined in Definition 4.6. The optimal $F_\theta$ that minimizes $\mathcal{L}(\theta)$ satisfies:*

$$F_\theta(z, f_x, t) = GG^\top(GG^\dagger z - f_y) + (\sigma_{\min} - 1)z.$$

## 5 Approximation

In this section, we utilize the approximation ability of the transformer-based neural networks, especially, Diffusion Transformer (DiT). First, in Section 5.1, we present the DiT backbone, a widely adopted model in empirical research. Next, we introduce the main theorem in Section 5.2, which provides an approximation result and establishes an upper bound on the error.

### 5.1 Diffusion Transformer (DiT)

Diffusion Transformer (Peebles & Xie, 2023) introduces a strategy where Transformers (Vaswani et al., 2017) serve as the core architecture for Diffusion Models (Ho et al., 2020; Song et al., 2020a). In particular, each Transformer block comprises a multi-head self-attention module and a feed-forward component, both of which include skip connections. In this paper, Transformer networks with positional encoding $E \in \mathbb{R}^{L \times d}$ is used in the analysis. For the formal definitions, please refer to Section A.2. We take a Transformer network consisting $K$ blocks and positional encoding as an example:

**Example 5.1.** *We here give an example for the sequence-to-sequence mapping $f_\mathcal{T}$ in Definition A.7: Denote $K$ as the number of layers in some transformer network. For an input matrix $X \in \mathbb{R}^{L \times d}$, we use $E \in \mathbb{R}^{L \times d}$ to denote the positional encoding, we then define:*

$$f_\mathcal{T}(X) = \mathsf{TF}_{(K)}^{h,m,r} \circ \cdots \circ \mathsf{TF}_{(1)}^{h,m,r}(X + E).$$

### 5.2 Main Theorem I: Approximation

We first present the universal approximation theorem for transformer-based models and utilize it as a lemma to establish our main theorem..

**Lemma 5.2** (Theorem 2 of (Yun et al., 2020)). *Let $\epsilon > 0$ and let $\mathcal{F}_{\mathrm{PE}}$ be the function class consisting all continuous permutation equivariant functions with compact support that $\mathbb{R}^{L \times d} \to \mathbb{R}^{L \times d}$. For any $f, g : \mathbb{R}^{L \times d} \to \mathbb{R}^{L \times d}$ be two different functions, we can show that for any given $f \in \mathcal{F}_{\mathrm{PE}}$, there exists a Transformer $g \in \mathcal{T}^{h,m,r}$ such that $\|f(X) - g(X)\|_2 \le \epsilon, \forall X \in \mathbb{R}^{L \times d}$.*

Before we state the approximation theorem, we define a reshaped layer that transforms concatenated input in flow matching into a length-fixed sequence of vectors.

**Definition 5.3** (DiT reshape layer). *Let $R : \mathbb{R}^{N+1} \to \mathbb{R}^{n \times d}$ be a reshape layer that transforms the $(N+1)$-dimensional input vector into a $n \times d$ matrix.*

Therefore, in the following, we give the theorem utilizing DiT to minimize the training objective $\mathcal{L}(\theta)$ to arbitrary error.

**Theorem 5.4** (Informal version of Theorem C.1). *Let the DiT reshape layer $R$ be defined in Definition 5.3. There exists a transformer network $f_{\mathcal{T}} \in \mathcal{T}_P^{2,1,4}$ defining function $F_\theta(z, f_x, t) := f_{\mathcal{T}}(R([z^\top, f_x^\top, t]^\top))$ with parameters $\theta$ that satisfies $\mathcal{L}(\theta) \le \epsilon$ for any error $\epsilon > 0$.*

# 6 GENERALIZATION

This section establishes generalization guarantees for the transformer-based sampling algorithm by combining analytical tools and convergence results. Section 6.1 introduces an error bound $\epsilon_1$ for the regularized function $\widehat{F}$ under noisy sampling, while Section 6.2 leverages these bounds to prove the transformer network's asymptotic generalization error $\epsilon_0 + \epsilon_1$, connecting algorithmic stability with approximation-theoretic guarantees.

## 6.1 BASIC TOOLS

We define another regularized function $\widehat{F}(f_x) := M(\mathcal{I}_y)P(M(\mathcal{I}_x)P)^\dagger f_x$, then we have:

**Lemma 6.1** (Informal version of Lemma C.2). *Let $\delta \in (0, 0.1)$. For any in-distribution (ID) data $f^8 \in H^s(\mathcal{X})$ be defined in Definition 3.1 and its corresponding sample $f \in \mathbb{D}$, we define:*

$$\epsilon_1 := O\Big( \frac{\sqrt{\frac{v}{\delta}} \exp(O(nN))}{\lambda} + \frac{N^{1.5}L}{\sqrt{n}} \Big)^2.$$

*where $v$ is the variance of noise under Definition 3.1. Then with a probability at least $1 - \delta$, we have*

$$\mathbb{E}_{f \in \mathcal{D}}[\|\widehat{F}(f_x) - f_y\|_2^2] \le \epsilon_1.$$

## 6.2 MAIN THEOREM II: GENERALIZATION

We present our generalization result as follows:

**Theorem 6.2.** *Denote the failure probability $\delta \in (0, 0.1)$ and an arbitrary error $\epsilon_0 > 0$. There exists a transformer network $f_{\mathcal{T}} \in \mathcal{T}_P^{2,1,4}$ defining function $F_\theta(z, f_x, t) := f_{\mathcal{T}}(R([z^\top, f_x^\top, t]^\top))$ with parameters $\theta$ that satisfies: for any in-distribution (ID) data $f \in \mathcal{D}$ and its corresponding signal $g : \mathbb{R}_{\ge 0} \to \mathbb{R}$, we sample new data $\widetilde{f} := [g(\tau \cdot \Delta) + \xi_\tau]$, where $\Delta$ is the sample step size. We denote $x_1$ as the output of Algorithm 1 with $T$ steps. Then with a probability at least $1 - \delta$, we have:*

$$\lim_{T \to +\infty} \mathbb{E}_{x_0 \sim \mathcal{N}(0, I_{N_y}), f \in \mathcal{D}}[\|x_1 - \widetilde{f}_y\|_2^2] \le \epsilon_0 + \epsilon_1$$

*Proof.* This proof combines from Lemma 6.1 and other proofs are similar with the ones in Theorem 5.4 since we suggest the transformer network to represent the function $\widehat{F}$. $\qquad\square$

# 7 EFFICIENCY

Here in this section, we consider the sampling efficiency problem of the vanilla sampling process of flow matching for time series generation (Algorithm 1). This section analyzes the convergence properties of the sampling algorithm through gradient descent, establishing error decrease and overall efficiency. Section 7.1 analyzes the error decrease per iteration by establishing gradient descent updates and key properties including Lipschitz smoothness, unbiased updates, and update norms, while Section 7.3 establishes the overall convergence rate of the algorithm by bounding the minimum expected gradient norm across iterations, demonstrating efficiency under chosen parameters.

Moreover, in Section 7.2, we present the sampling algorithm for generating time series to review the sampling process of flow matching as an optimization process, utilizing the previously defined conditional flow and training objective.

## 7.1 ERROR DECREASE

**Gradient descent with respect to some objective.** As we define the polynomial regularization in Definition 4.5, we claim that Algorithm 1 implements a first-order gradient descent to some implicit parameter, we denote it as $w : [T] \to \mathbb{R}^n$. Formally, we define $w$ as

**Definition 7.1** (Implicit parameter $w$). *Let $P$ be defined in Definition 4.1. Let $f_y$ be defined in Definition 3.5. We denote the implicit parameter $w$ as $w : [T] \to \mathbb{R}^n$, i.e., $w_t \in \mathbb{R}^n$ for time step $t$. Particularly, we define $w_0 := P^\dagger x_0$ as the initialization and $w^* := P^\dagger f_y$ as the optimal solution.*

Besides, we use the metric that measures the square $\ell_2$ norm of the difference between the current sampling result $x_{\frac{t}{T}}$ and the ground truth. Formally, we define the metric as follows:

**Definition 7.2** (Metric). *Let $w$ be defined in Definition 7.1. Let $P$ be defined in Definition 4.1. Let $f$ and $f_y$ be defined in Definition 3.5. We define the metric $u : \mathbb{R}^n \to \mathbb{R}$ as $u(w_t) := \mathbb{E}_{f \in \mathcal{D}}[\|Pw_t - f_y\|_2^2]$.*

Then the update is given by:

**Definition 7.3** (Update Rule). *Let $w$ be defined in Definition 7.1. Let $P$ be defined in Definition 4.1. Let $F_\theta : \mathbb{R}^{N_y} \times \mathbb{R}^{N_x} \times [0,T] \to \mathbb{R}^{N_y}$ be the model with parameter $\theta$. Let $\sigma_t$ be the time-dependent standard deviation. Let $f_x$ and $f_y$ be defined in Definition 3.5. Let $z \sim \mathcal{N}(0, I_{N_y})$ be the sample. We use $\Delta w_t$ to denote the weight adjustment, which is defined as $\Delta w_{t-1} := P^\dagger \left( T \cdot F_\theta(Pw_{t-1}, f_x, \frac{t-1}{T}) + z \cdot \sigma_{\frac{t}{T}}(f) \right)$. In each iteration, we update the parameter as $w_t = w_{t-1} - \Delta w_{t-1}$.*

**Lemma 7.4.** *Let $w$ be defined in Definition 7.1. Let $\alpha$ be the constant in Definition 4.5. Let $P$ be defined in Definition 4.1. Let $F_\theta : \mathbb{R}^{N_y} \times \mathbb{R}^{N_x} \times [0,T] \to \mathbb{R}^{N_y}$ be the model with parameter $\theta$. Let $f_x$ and $f_y$ be defined in Definition 3.5. Let $G$ be defined in Definition 4.4. We can show that $\|P^\dagger F_\theta(Pw_t, f_x, \frac{t}{T}) - \alpha G^\top(Gw_t - f_y)\|_2^2 \le \epsilon_0$, where $\epsilon_0 > 0$ is an arbitrary positive error.*

*Proof.* This result follows from Lemma 5.2. $\qquad\square$

First, we give the some tools in helping the analysis as follows:

**Lemma 7.5** (Informal version of Lemma C.3). *Let $w$ be defined in Definition 7.1. Let $t, t' \in [0, T]$ be two different time step. Let $u(w_t)$ be defined in Definition 7.2. Let $\lambda := \lambda_{\min}(P) > 0$. Let $\alpha$ be the constant in Definition 4.5. Let $G$ be defined in Definition 4.4. Let $\sigma_t$ be defined in Definition 4.6. Let $\Delta w_t$ be defined in Definition 7.3. Let $f$ be defined in Definition 3.5. Then we have*

- **Lipschitz-smooth.** $\forall w_t, w_{t'} \in \mathbb{R}^n$, $\|\nabla_{w_t} u(w_t) - \nabla_{w_{t'}} u(w_{t'})\|_2 \le \frac{n \exp(O(nN))}{\lambda} \|w_t - w_{t'}\|_2$.

- **Unbiased update.** $\mathbb{E}[\Delta w_t] = \alpha T \cdot \mathbb{E}[\nabla_{w_t} u(w_t)]$.

- **Update norm.** $\mathbb{E}[\|\Delta w_t\|_2^2] = \alpha^2 T^2 \cdot \mathbb{E}[\|\nabla_{w_t} u(w_t)\|_2^2] + n \cdot \sigma_{\frac{t}{T}}(f)$.

Thus, we prove the expectation of error decrease of sampling at each step, as we state below:

**Lemma 7.6** (Informal version of Lemma C.4). *We define $L_1 := n \cdot \frac{\exp(O(nN))}{\lambda}$. Let $w$ be defined in Definition 7.1. Let $u(w_t)$ be defined in Definition 7.2. Let $\alpha$ be the constant in Definition 4.5. Let $\sigma_t(f)$ be defined in Definition 4.6. Let $f$ be defined in Definition 3.5. For each step $t \in [T]$, we have:*

$$\mathbb{E}[u(w_t)] \le \mathbb{E}[u(w_{t-1})] + (\frac{L_1}{2}\alpha^2 T^2 - \alpha T) \mathbb{E}[\|\nabla_{w_{t-1}} u(w_{t-1})\|_2^2] + \frac{L_1 n}{2}\sigma_{\frac{t-1}{T}}(f)$$

## 7.2 Sampling Algorithm

Now we review the algorithm form of the sampling process of flow matching for time series generation in Algorithm 1.

---

**Algorithm 1** Recall the sampling process of flow matching for time series generation

---

**Input:** Time series $f_x \in \mathbb{R}^{N_x}$, sample steps $T > 0$
**Output:** Predictive time series $x_1 \in \mathbb{R}^{N_y}$
1: **procedure** SAMPLING($f_x$)
2:     Sample the initial Gaussian noise $x_0 \in \mathcal{N}(0, I_{N_y})$
3:     **for** $t \in [T]$ **do**
4:         If $t > 1$, sample $z \sim \mathcal{N}(0, I_y)$; else, $z = \mathbf{0_{N_y}}$
5:         Update $x_{\frac{t}{T}} \leftarrow x_{\frac{t-1}{T}} - T \cdot F_\theta(x_{\frac{t-1}{T}}, f_x, \frac{t-1}{T})$
6:         Update $x_{\frac{t}{T}} \leftarrow x_{\frac{t}{T}} - (1 - (1 - \sigma_{\min})\frac{t}{T}) \cdot z$
7:     **end for**
8:     **return** $x_1$
9: **end procedure**

---

## 7.3 Main Theorem III: Convergence

Here, we demonstrate the efficiency of the sample process below:

**Theorem 7.7** (Informal version of Theorem C.5). *Let $w$ be defined in Definition 7.1. Let $u(w_t)$ be defined in Definition 7.2. Let $\delta \in (0, 0.1)$. Denote the failure probability $1 - \delta$. For error $\epsilon > 0$, we choose $T = \widetilde{O}(\sqrt{N/(L_1 \alpha \epsilon)})$, then with a probability at least $1 - \delta$, we have:* $\min_{t \in [T]} \mathbb{E}[\|\nabla_{w_t} u(w_t)\|_2^2] \leq \epsilon$.

# 8 Limitation

Our work is intentionally theoretical, and we do not provide empirical results. The goal is to establish rigorous approximation, generalization, and efficiency guarantees for generative models whose empirical success has already been demonstrated in prior studies. Our guarantees rely on assumptions tailored to continuous, regression-style time-series data, including the noise model and time-indexed sampling structure in Section 3. Extending the analysis to discrete modalities, would require redefining the underlying noise model, flow dynamics, and regularity assumptions. Exploring these broader settings and conducting empirical studies to complement our theory are promising directions for future work.

# 9 Conclusion

This paper establishes a theoretical framework for understanding flow-based generative models in time series analysis, addressing the critical gap between empirical success and theoretical foundations. By integrating polynomial regularization into the flow matching objective, we demonstrate that transformer-based architectures can achieve provable approximation, generalization, and convergence guarantees. Our analysis reveals three key insights: (1) Diffusion Transformers universally approximate the optimal flow matching objective, (2) polynomial regularization enables generalization bounds combining approximation errors and noise tolerance, and (3) the sampling process exhibits gradient descent-like convergence under Lipschitz smoothness conditions. These results provide the first end-to-end theoretical justification for modern time series generation paradigms, confirming that architectural choices like DiT and training strategies like flow matching jointly enable both expressivity and stability. Future work could extend this framework to non-Gaussian noise settings and investigate the tightness of our polynomial-dependent error bounds. More broadly, our methodology opens new avenues for theoretically grounding other temporal generative models while maintaining alignment with practical implementations.

## ETHIC STATEMENT

This paper does not involve human subjects, personally identifiable data, or sensitive applications. We do not foresee direct ethical risks. We follow the ICLR Code of Ethics and affirm that all aspects of this research comply with the principles of fairness, transparency, and integrity.

## REPRODUCIBILITY STATEMENT

We ensure reproducibility of our theoretical results by including all formal assumptions, definitions, and complete proofs in the appendix. The main text states each theorem clearly and refers to the detailed proofs. No external data or software is required.

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

# Appendix

**Roadmap.** Section A presents some useful preliminary definitions and lemmas. Section B presents the optimal solution of the neural network. Section C presents the missing proof of our main results. Section D states the impact of the paper.

## A PRELIMINARY

### A.1 BASIC TOOLS

**Lemma A.1.** *Following Definition 3.1, we have two properties that hold as follows:*

- *Property 1. $f^*$ is Lipchitz smooth, we denote the smoothness as $L_0$.*

- *Property 2. Denote the failure probability $\delta \in (0, 0.1)$, then with a probability at least $1 - \delta$, we have $|f^*(t) + \xi| \leq \sqrt{\frac{v}{\delta}}$.*

*Proof.* **Proof of Property 1.** Once $s \geq d/2 + 1$ for the function in $H^s(\Omega)$ and $d$ is the dimension of $\Omega$, the whole $H^s(\Omega)$ is embedded by $C^1$, therefore, this property is trivial since $d = 1$ and $s \geq 1.5$.

**Proof of Property 2.** Here we state Chebyshev's inequalities: For a random variable $X$ with finite mean $\mu$ and variance $\sigma^2$, then for any $t > 0$, we have:

$$\Pr[|X - \mu| \geq t\sigma] \leq \frac{1}{t^2}.$$

We thus apply Chebyshev's inequality to $\xi$ to obtain the result. $\qquad\square$

**Lemma A.2.** *For a PD matrix $A \in \mathbb{R}^{d_1 \times d_2}$ with a positive minimum singular value $\lambda_{\min}(A) > 0$, the infinite norm of its pseudoinverse matrix $A^\dagger$ is given by:*

$$\|A^\dagger\| \leq \frac{1}{\lambda_{\min}(A)}.$$

*Proof.* We have:

$$\begin{aligned}
\|A^\dagger\| &= \|(U\Sigma V)^\dagger\| \\
&= \|V^\top \Sigma^\dagger U^\top\| \\
&= \|\Sigma^\dagger\| \\
&\leq \frac{1}{\lambda_{\min}(A)}
\end{aligned}$$

where the first step follows from the svd of $A = U\Sigma V$, the second step follows from simple algebras, the third step follows from $U, V$ are orthogonal (and square) matrices, the last step follows from the definitions of the spectral norm and $\Sigma$ is a diagonal matrix of singular values. $\qquad\square$

**Lemma A.3.** *For two matrices $A, B \in \mathbb{R}^{d_1 \times d_2}$, we have:*

$$\|A^\dagger - B^\dagger\| \leq \max\{\|A^\dagger\|^2, \|B^\dagger\|^2\} \cdot \|A - B\|.$$

*Proof.* We have:

$$\begin{aligned}
\|A^\dagger - B^\dagger\| &\leq \|A^\dagger\| \cdot \|I_{d_1} - AB^\dagger\| \\
&\leq \|A^\dagger\|\|B^\dagger\| \cdot \|A - B\| \\
&\leq \max\{\|A^\dagger\|^2, \|B^\dagger\|^2\} \cdot \|A - B\|
\end{aligned}$$

where these steps follow from simple algebras and $A^\dagger A \approx I_{d_1}$ $\qquad\square$

## A.2 DiT

We first define the multi-head self-attention:

**Definition A.4** (Multi-head self-attention). *Given $h$-heads query, key, value and output projection weights $W_Q^i, W_K^i, W_V^i, W_O^i \in \mathbb{R}^{d \times m}$ with each weight is a $d \times m$ shape matrix, for an input matrix $X \in \mathbb{R}^{L \times d}$, we define a multi-head self-attention $\mathsf{Attn} : \mathbb{R}^{L \times d} \to \mathbb{R}^{L \times d}$ as follows:*

$$\mathsf{Attn}(X) := \sum_{i=1}^{h} \mathsf{Softmax}(X W_Q^i {W_K^i}^\top X^\top) \cdot X W_V^i {W_O^i}^\top + X.$$

A feed-forward layer transforms input data by applying linear projections, a non-linear activation function, and residual connections, which is defined as follows:

**Definition A.5** (Feed-forward). *Given two projection weights $W_1, W_2 \in \mathbb{R}^{d \times r}$ and two bias vectors $b_1 \in \mathbb{R}^r$ and $b_2 \in \mathbb{R}^d$, for an input matrix $X \in \mathbb{R}^{L \times d}$, we define a feed-forward computation $\mathsf{FF} : \mathbb{R}^{L \times d} \to \mathbb{R}^{L \times d}$ follows:*

$$\mathsf{FF}(X) := \phi(X W_1 + \mathbf{1}_L b_1^\top) \cdot W_2^\top + \mathbf{1}_L b_2^\top + X.$$

*Here, $\phi$ is an activation function and usually be considered as ReLU.*

We denote a Transformer block as $\mathsf{TF}^{h,m,r} : \mathbb{R}^{L \times d} \to \mathbb{R}^{L \times d}$, where $h$ is the count of attention heads, $m$ specifies the head dimension within the self-attention mechanism, and $r$ is the hidden size in the feed-forward layer. Building on multi-head self-attention and the feed-forward layer, we define the transformer block as follows:

**Definition A.6** (Transformer block). *Let multi-head self-attention and feed-forward neural network be defined in Definition A.4 and Definition A.5 respectively. Formally, for an input matrix $X \in \mathbb{R}^{L \times d}$, we define the Transformer block $\mathsf{TF}^{h,m,r} : \mathbb{R}^{L \times d} \to \mathbb{R}^{L \times d}$:*

$$\mathsf{TF}^{h,m,r}(X) := \mathsf{FF} \circ \mathsf{Attn}(X)$$

We define the Transformer network as the composition of Transformer blocks:

**Definition A.7** (Complete transformer network). *We consider a transformer network as a composition of a transformer block (Definition A.6) with model weight $\theta^{h,m,r}$, which is:*

$$\mathcal{T}^{h,m,r} := \{\mathcal{F} : \mathbb{R}^{L \times d} \to \mathbb{R}^{L \times d} \mid \mathcal{F} \text{ is a composition of Transformer blocks } \mathsf{TF}_{\theta^{h,m,r}}\text{'s}$$
$$\text{with positional embedding } E \in \mathbb{R}^{L \times d}\}$$

# B CLOSE FORM OF OPTIMAL SOLUTION

We then provide the closed-form solution for $F_\theta$ that achieves the minimum of $\mathcal{L}(\theta)$ as follows:

**Theorem B.1** (Formal version of Theorem 4.9). *If the following conditions hold:*

- *Let $\mathcal{L}(\theta)$ be defined in Definition 4.8.*

- *Let $z \sim \mathcal{N}(0, I_{N_y})$.*

- *Let $t \sim \mathsf{Uniform}[0, T]$.*

- *Let $G$ be defined in Definition 4.4.*

- *Let $f_x, f_y$ be defined in Definition 3.5.*

- *Let $\sigma_{\min}$ be defined in Definition 4.6.*

*The optimal $F_\theta$ that minimizes $\mathcal{L}(\theta)$ satisfies:*

$$F_\theta(z, f_x, t) = GG^\top\big(GG^\dagger z - f_y\big) + (\sigma_{\min} - 1)\, z.$$

*Proof.* Observe that

$$\psi_t'(f) = \mu_t'(f) + \sigma_t'(f) \cdot z$$
$$= GG^\top(GG^\dagger \psi_t(f) - f_y) + (\sigma_{\min} - 1)z,$$

where the initial step follows from the construction and definition of $\psi_t(f)$, and the subsequent step is due to Definition 4.5. Substituting $\psi_t(f)$ with $z$ completes the derivation. $\square$

## C  MISSING PROOFS

In Section C.1, we present the missing proof in Section 5. In Section C.2, we present the missing proof in Section 6. In Section C.3, we present the missing proof in Section 7.

### C.1  APPROXIMATION

**Theorem C.1** (Formal version of Theorem 5.4). *If the following conditions hold:*

- *Let the DiT reshape layer $R$ be defined in Definition 5.3.*

*Then there exists a transformer network $f_\mathcal{T} \in \mathcal{T}_P^{2,1,4}$ defining function $F_\theta(z, f_x, t) := f_\mathcal{T}(R([z^\top, f_x^\top, t]^\top))$ with parameters $\theta$ that satisfies $\mathcal{L}(\theta) \leq \epsilon$ for any error $\epsilon > 0$.*

*Proof.* Choose $L = 1$ for $R(\cdot)$, we define:

$$f_\mathcal{T}^*([z^\top, f_x^\top, t]^\top) := GG^\top(GG^\dagger z - F^*(f_x)) + (\sigma_{\min} - 1)z.$$

Then, following Lemma 5.2, there exists a transformer network $f_\mathcal{T} \in \mathcal{T}_P^{2,1,4}$ that satisfies (arbitrary error $\epsilon > 0$):

$$\|f_\mathcal{T}(R([z^\top, f_x^\top, t]^\top)) - f_\mathcal{T}^*([z^\top, f_x^\top, t]^\top)\|_2 \leq \epsilon.$$

Since $\|P\|_\infty \leq \exp(O(nN))$, we have $\|GG^\top\|_2 \leq N\exp(O(nN))$, scaling $\epsilon \leq \frac{\epsilon_0}{N\exp(O(nN))}$ could directly achieve the theorem result. $\square$

### C.2  GENERALIZATION

**Lemma C.2** (Formal version of Lemma 6.1). *If the following conditions hold:*

- *Let $\delta \in (0, 0.1)$.*

- *Let $\epsilon_1 := O\left(\frac{\sqrt{\frac{v}{\delta}}\exp(O(nN))}{\lambda} + \frac{N^{1.5}L}{\sqrt{n}}\right)^2$ be the error bound, where $v$ is the variance of noise under Definition 3.1.*

- *Let in-distribution (ID) data $f \in \mathcal{D}$ be defined in Definition 3.5.*

- *Let $g : \mathbb{R}_{\geq 0} \to \mathbb{R}$ be the corresponding signal of $f$.*

- *Let $\widetilde{f} := [g(\tau \cdot \Delta) + \xi_\tau]$ be a new sampled data, where $\Delta$ is the sample step size.*

*Then with a probability at least $1 - \delta$, we have*

$$\mathbb{E}_{f \in \mathcal{D}}[\|\widehat{F}(\widetilde{f}_x) - \widetilde{f}_y\|_2^2] \leq \epsilon_1.$$

*Proof.* We have:

$$\mathbb{E}_{f \in \mathcal{D}}[\|\widehat{F}(\widetilde{f}_x) - \widetilde{f}_y\|_2] \leq \mathbb{E}_{f \in \mathcal{D}}[\|M(\mathcal{I}_y)P(M(\mathcal{I}_x)P)^\dagger \widetilde{f}_x - M(\mathcal{I}_y)PP^\dagger \widetilde{f}\|_2] + O(N^{1.5}L/\sqrt{n})$$

$$\leq \|P\| \cdot \mathbb{E}_{f \in \mathcal{D}}[\|(M(\mathcal{I}_x)P)^\dagger \widetilde{f}_x - P^\dagger \widetilde{f}\|_2] + O(N^{1.5}L/\sqrt{n})$$

$$\leq \|P\| \cdot \mathop{\mathbb{E}}_{f \in \mathcal{D}}[\|(M(\mathcal{I}_x)P)^\dagger - P^\dagger\| \cdot \|\widetilde{f}_x\|_2$$

$$+ \|P^\dagger\|\|M(\mathcal{I}_x)^\dagger \widetilde{f}_x - \widetilde{f}\|_2] + O(N^{1.5}L/\sqrt{n})$$

$$\leq \frac{\sqrt{v/\delta}\exp(O(nN))}{\lambda} + O(N^{1.5}L/\sqrt{n})$$

where the first step follows from the polynomial approximation (Lemma 4.3), the second step follows from Cauchy-Schwarz inequality, the third step follows from simple algebras and triangle inequality, and the last step follows from some simple calculations with Lemma A.2 and Lemma A.3. □

## C.3 EFFICIENCY

**Lemma C.3** (Formal version of Lemma 7.5). *If the following conditions hold:*

- *Let $w$ be defined in Definition 7.1.*

- *Let $t, t' \in [0, T]$ be two different time step.*

- *Let $u(w_t)$ be defined in Definition 7.2.*

- *Let $\lambda := \lambda_{\min}(P) > 0$.*

- *Let $\alpha$ be the constant in Definition 4.5.*

- *Let $G$ be defined in Definition 4.4.*

- *Let $\sigma_t$ be defined in Definition 4.6.*

- *Let $\Delta w_t$ be defined in Definition 7.3.*

- *Let $f$ be defined in Definition 3.5.*

*Then we have:*

- **Lipschitz-smooth.** $\forall w_t, w_{t'} \in \mathbb{R}^n$,

$$\|\nabla_{w_t} u(w_t) - \nabla_{w_{t'}} u(w_{t'})\|_2 \leq \frac{n\exp(O(nN))}{\lambda}\|w_t - w_{t'}\|_2.$$

- **Unbiased update.**

$$\mathbb{E}[\Delta w_t] = \alpha T \cdot \mathbb{E}[\nabla_{w_t} u(w_t)].$$

- **Update norm.**

$$\mathbb{E}[\|\Delta w_t\|_2^2] = \alpha^2 T^2 \cdot \mathbb{E}[\|\nabla_{w_t} u(w_t)\|_2^2] + n \cdot \sigma_{\frac{t}{T}}(f).$$

*Proof.* **Proof of gradient Lipschitz-smooth.** We have:

$$\|\nabla_{w_t} u(w_t) - \nabla_{w_{t'}} u(w_{t'})\|_2 = \|G^\top(Gw_t - \mathbb{E}[f_y]) - G^\top(Gw_{t'} - \mathbb{E}[f_y])\|_2$$

$$= \|G^\top(Gw_t - Gw_{t'})\|_2$$

$$\leq \|G^\top G\|_2 \cdot \|w_t - w_{t'}\|_2$$

$$\leq \frac{n\exp(O(nN))}{\lambda}\|w_t - w_{t'}\|_2,$$

where the first step follows from the derivation of $u(w)$, the second step follows from simple algebras, the third step follows from Cauchy-Schwarz inequality, the last step follows from $\|G\|_\infty \leq \exp(O(nN))$.

**Proof of unbiased update.** We have:

$$\mathbb{E}[\Delta w_t] = \mathbb{E}[P^\dagger\Big(TF(Pw_{t-1}, f_x, \frac{t-1}{T}) + \sigma_{\frac{t}{T}}(f)z\Big)]$$

$$= \alpha T G^\top (G w_t - \mathbb{E}[f_y])$$
$$= \alpha T \nabla_{w_t} u(w_t),$$

where the first step follows from Definition 7.3, the second step follows from $\mathbb{E}[z] = \mathbf{0}_d$, the last step follows from the derivation of $u(w)$.

**Proof of update norm.** We have:

$$\mathbb{E}[\|\Delta w_t\|_2^2] = \alpha^2 T^2 \, \mathbb{E}[\|\nabla_{w_t} u(w_t)\|_2^2] - \alpha T \, \mathbb{E}[\sigma_{\frac{t}{T}}(f) \langle \nabla_{w_t} u(w_t), z \rangle] + \mathbb{E}[\|\sigma_{\frac{t}{T}}(f) z\|_2^2]$$
$$= \alpha^2 T^2 \, \mathbb{E}[\|\nabla_{w_t} u(w_t)\|_2^2] + \mathbb{E}[\|\sigma_{\frac{t}{T}}(f) z\|_2^2]$$
$$= \alpha^2 T^2 \, \mathbb{E}[\|\nabla_{w_t} u(w_t)\|_2^2] + \sigma_{\frac{t}{T}}(f) n$$

where the first step follows from Definition 7.3, the second step follows from $\mathbb{E}[z] = \mathbf{0}_d$, the last step follows from $E[\|z\|_2^2] = n$ (the variance of Gaussian distribution). $\square$

**Lemma C.4** (Formal version of Lemma 7.6). *If the following conditions hold:*

- *We define $L_1 := n \cdot \frac{\exp(O(nN))}{\lambda}$.*

- *Let $w$ be defined in Definition 7.1.*

- *Let $u(w_t)$ be defined in Definition 7.2.*

- *Let $\alpha$ be the constant in Definition 4.5.*

- *Let $\sigma_t(f)$ be defined in Definition 4.6.*

- *Let $f$ be defined in Definition 3.5.*

*Then for each step $t \in [T]$, we have:*

$$\mathbb{E}[u(w_t)] \leq \mathbb{E}[u(w_{t-1})] + (\frac{L_1}{2}\alpha^2 T^2 - \alpha T) \, \mathbb{E}[\|\nabla_{w_{t-1}} u(w_{t-1})\|_2^2] + \frac{L_1 n}{2} \sigma_{\frac{t-1}{T}}(f)$$

*Proof.* We first give a common tool for proving convergence that is derived from Taylor expansion, such that:

$$u(w_t) \leq u(w_{t-1}) - \langle \nabla_{w_{t-1}} u(w_{t-1}), \Delta w_{t-1} \rangle + \frac{L_1}{2} \|\Delta w_{t-1}\|_2^2.$$

Next, taking expectation to the whole equation, we can get:

$$\mathbb{E}[u(w_t)] \leq \mathbb{E}[u(w_{t-1}) - \langle \nabla_{w_{t-1}} u(w_{t-1}), \Delta w_{t-1} \rangle + \frac{L_1}{2} \|\Delta w_{t-1}\|_2^2]$$
$$= \mathbb{E}[u(w_{t-1})] - \alpha T \, \mathbb{E}[\|\nabla_{w_{t-1}} u(w_{t-1})\|_2^2]$$
$$+ \frac{L_1}{2}(\alpha^2 T^2 \, \mathbb{E}[\|\nabla_{w_{t-1}} u(w_{t-1})\|_2^2] + \sigma_{\frac{t-1}{T}}(f) n)$$
$$\leq \mathbb{E}[u(w_{t-1})] + (\frac{L_1}{2}\alpha^2 T^2 - \alpha T) \, \mathbb{E}[\|\nabla_{w_{t-1}} u(w_{t-1})\|_2^2] + \frac{L_1}{2} \sigma_{\frac{t-1}{T}}(f) n$$

where the second step follows from Lemma 7.5, the third step follows from some simple algebras. $\square$

**Theorem C.5** (Formal version of Theorem 7.7). *If the following conditions hold:*

- *Let $w$ be defined in Definition 7.1.*

- *Let $u(w_t)$ be defined in Definition 7.2.*

- *Let $\delta \in (0, 0.1)$.*

- *Let $T = \widetilde{O}(\sqrt{N/(L_1 \alpha \epsilon)})$.*

- *Let $1 - \delta$ be the failure probability.*

*Then for error $\epsilon > 0$, with a probability at least $1 - \delta$, we have:*

$$\min_{t \in [T]} \mathbb{E}[\|\nabla_{w_t} u(w_t)\|_2^2] \leq \epsilon$$

*Proof.* We have:

$$\min_{t \in [T]} \mathbb{E}[\|\nabla_{w_t} u(w_t)\|_2^2] \leq \frac{1}{T} \sum_{t=1}^{T} \mathbb{E}[\|\nabla_{w_t} u(w_t)\|_2^2]$$

$$\leq \frac{1}{\alpha T^2 (0.5 L_1 \alpha T - 1)} \sum_{t=1}^{T} \mathbb{E}[u(w_{t-1})] - \mathbb{E}[u(w_t)] + \frac{L_1 n}{2} \sigma_{\frac{t}{T}}(f)$$

$$\leq \frac{1}{\alpha T^2 (0.5 L_1 \alpha T - 1)} \sum_{t=1}^{T} \mathbb{E}[u(w_{t-1})] - \mathbb{E}[u(w_t)] + \frac{L_1 n}{2}$$

$$\leq \frac{1}{\alpha T^2 (0.5 L_1 \alpha T - 1)} \left( \mathbb{E}[u(w_0)] - \mathbb{E}[u(w_T)] + \frac{L_1 n T}{2} \right) \qquad (1)$$

where the first step follows from the minimum is always smaller than the average, the second step follows from Lemma 7.6, the third step follows from $\sigma_t(f) \leq 1$, the fourth step follows from simple algebras.

For the term $\mathbb{E}[u(w_0)] - \mathbb{E}[u(w_T)]$ in Eq. (1), we can show that

$$\mathbb{E}[u(w_0)] - \mathbb{E}[u(w_T)] \leq \mathbb{E}[u(w_0)]$$
$$\leq O(N \log(N/\delta)) \qquad (2)$$

where the first step follows from $u(w) \geq 0$ for any $w \in \mathbb{R}^n$, the second step follows from Gaussian tail bound and the upper bound on $f_y$.

Combine Eq. (1) and Eq. (2), we can show that

$$\min_{t \in [T]} \mathbb{E}[\|\nabla_{w_t} u(w_t)\|_2^2] \leq \epsilon$$

which follows from $T = \widetilde{O}(\sqrt{N/(L_1 \alpha \epsilon)})$. $\qquad \square$

## D   IMPACT STATEMENTS

This research shows why flow matching performs good on time series forecasting, which map intricate relationships between many items. This could lead to more powerful AI tools for solving complex problems. As this work is theoretical and focuses on the capability of these models, we don't foresee direct negative societal impacts.

## LLM USAGE DISCLOSURE

LLMs were used only to polish language, such as grammar and wording. These models did not contribute to idea creation or writing, and the authors take full responsibility for this paper's content.

