# OpenReview forum: "Provable Guarantees for Flow-Based Generative Models in Time Series"
_ICLR.cc/2026/Conference — Submitted to ICLR 2026_

### Official Review · Reviewer_6YoR · 2025-10-24

**Soundness:** 3
**Presentation:** 3
**Contribution:** 3
**Rating:** 6
**Confidence:** 2

**Summary:**

This paper presents a unified theoretical framework for flow-based generative models specifically tailored for time-series generation and forecasting, providing three key classes of theoretical guarantees. Firstly, for Approximation, the authors demonstrate, by leveraging the universal approximation properties of Diffusion Transformers (DiT), that the model class can approximate the optimal conditional flow with arbitrary precision. Secondly, concerning Generalization, the introduction of a polynomial-basis regularization for the conditional flow results in an explicit upper bound on the generalization error, which is established under a noisy time-series model. Finally, in terms of Efficiency, the paper establishes convergence rates for the sampling process by casting it as a first-order optimization procedure under specific smoothness and regularity assumptions. Overall, the work's primary ambition is to be the first to deliver a comprehensive, end-to-end theoretical justification, covering approximation, generalization, and sampling convergence, for modern flow-based time-series models.

**Strengths:**

1. **Motivation.** The paper targets a well-defined and important gap: the theory of generative models for time series lags behind rapid empirical progress. Clarifying approximation, generalization, and convergence is valuable.

2. **End-to-end scope.** Addressing three pillars, expressivity, generalization, and sampling efficiency, in a single framework is ambitious and conceptually clean.

3. **Theoretical insights.** While some technical subtleties are intricate, the insights are interesting and broadly useful for understanding the capabilities and limitations of flow-based models. This line of work can help ground future scaling of time-series generative algorithms.

**Weaknesses:**

1. **Lack of empirical validation.** No experiments are provided to indicate whether the bounds are numerically valid or to illustrate the effect of polynomial regularization. Even a small synthetic study could substantially improve clarity and persuasiveness.

2. **Incrementality in Section 5.** From a non-expert perspective in approximation theory, the DiT-based universality result may read as an application of known transformer approximation theorems rather than a fundamentally new approximation insight specific to this setting.

3. **Single-dataset formalism.** The guarantees are presented for a single-distribution (single-dataset) setup. In the era of large models trained across multiple datasets, it would be helpful to discuss limitations or extensions when the model must handle mixture distributions or dataset shifts; the current framework does not directly answer these multi-dataset questions, although this does not diminish the novelty of the presented results.

**Questions:**

- **Notation in Lemma 6.1.** Can the author please clarify the definition of `\tilde{f}` and how it differs from the standard `f` ?

- **Scope beyond time series.** Are the results inherently tied to a regression-style time-series setting, or can the analysis (with adjusted assumptions) extend to other modalities/tasks (e.g., language modeling with logistic outputs)? A brief, intuitive paragraph detailing the differences and limitation could improve the paper quality.

- **Polynomial regularization in practice.** Is the polynomial-basis regularization intended primarily as a proof device, or do the authors advocate its practical use? If the latter, guidance on basis selection, order choice, and expected computational overhead would be valuable as well as empirical evidence as mentioned in the weaknesses.

- **Task scope.** Do the guarantees apply only to forecasting/imputation, or do they extend to unconditional generation ?

---

> ### Author Response · Authors · 2025-11-30
>
> We are very grateful for your positive and insightful comments. These comments are invaluable for improving our manuscript. Below, we want to respond to your questions.
>
> ### Weakness 1
> You note that the paper does not include empirical validation. We want to acknowledge that our submission is intentionally focused on developing a rigorous end-to-end theoretical framework for flow-based generative modeling. As we mentioned in the global response and limitation section in our revised version, the absence of empirical results in our work is intentional, and our focus in this paper is articulating the theoretical foundations for diffusion and flow based time series models, rather than duplicating empirical evidence already well established.
>
> ### Weakness 2
> Your observation is accurate that the universality result used in Section 5 relies on established transformer approximation theorems. Our unique contribution in that section is not to introduce a new approximation theorem, but to integrate known universality properties into the flow-matching framework and demonstrate that they yield end-to-end approximation guarantees for conditional time-series flows. This connection is specific to the setting of flow-based generative models for time series and enables the subsequent generalization and efficiency results.
>
> ### Weakness 3
> We fully agree that this can be meaningful extensions of our current theoretical scope. Extending our analysis to settings involving distribution mixtures, dataset shifts, or multitask training would require re-examining assumptions in both the generalization analysis and the noisy sampling model. This could be our future direction rather than current needs. We would explicitly acknowledge this in the revised version to avoid any misunderstanding about the generality of our results.
>
> ### Question 1
> Thank you for your careful reading, it is a typo and we have fixed it in our revision.
>
> ### Question 2
> Our current theoretical guarantees rely on assumptions tailored to time-series structures, particularly the noisy sampling model, the conditional target form, and the polynomial regularization design. While the techniques may inspire future adaptation, the guarantees we present are not claimed to apply outside time-series forecasting under the defined assumptions. We have added a limitation section in our paper to discuss this.
>
> ### Question 3
> Our analysis is developed specifically for conditional time-series generation, where the target distribution is continuous and supervised, and where the flow-matching objective operates over trajectories indexed by time. This setting naturally leads to assumptions such as a regression-style output, additive noise structure, and polynomial regularity conditions that match the smoothness of continuous-valued time-series data.
>
> Extending the theory to other modalities—for example, language modeling with discrete tokens or logistic outputs—would require reestablishing several fundamental components of the analysis, including:
>
> 1. The definition of the conditional flow, which in discrete domains would no longer evolve over a continuous Euclidean space.
> 2. The approximation assumptions, which currently rely on the smoothness of real-valued functions.
> 3. The generalization argument, which uses polynomial-based regularization tailored to continuous function classes.
>
> While the high-level ideas of flow approximation and sampling-as-optimization may inspire similar reasoning beyond time series, the guarantees proven in this paper are not claimed to extend directly to other modalities without substantial modification of the underlying assumptions.
>
> ### Question 4
> No, the guarantees do not extend to unconditional generation, the model still requires a valid input. Extending these results to unconditional generation would require different structural assumptions and is beyond the scope of the guarantees we provide.
>
> We appreciate your detailed reading and constructive suggestions.

---

### Official Review · Reviewer_PYku · 2025-10-29

**Soundness:** 3
**Presentation:** 3
**Contribution:** 2
**Rating:** 6
**Confidence:** 2

**Summary:**

This paper provides the first theoretical framework analyzing flow-based generative models for time series forecasting (TSF) from approximation, generalization, and efficiency perspectives.  This work aims to provide rigorous theoretical guarantees for understanding how flow-based models achieve approximation and generalization in TSF tasks.

**Strengths:**

1. The paper tackles a genuine need for theoretical understanding of generative models in TSF. While empirical success has been demonstrated, the lack of theoretical guarantees for approximation, generalization, and efficiency is a significant limitation that this work attempts to address.

2.  The framework covers three fundamental aspects (approximation, generalization, efficiency) providing a holistic theoretical treatment rather than focusing on a single dimension, which is valuable for complete understanding.

3.  Introducing polynomial-based regularization to bound generalization error is a concrete, actionable contribution that bridges theory with potential practical implementation.

**Weaknesses:**

1.   The paper appears to be purely theoretical without experiments validating the theoretical predictions. Do the approximation bounds, generalization bounds, and convergence rates hold in practice? Without empirical validation, the practical relevance of the theory is unclear.

2. The paper claims convergence to "arbitrary error" but doesn't discuss whether the bounds are tight or loose. Are the theoretical guarantees practically meaningful, or do they only hold asymptotically with unrealistic resource requirements?

I must note that I am not deeply familiar with the theoretical analysis of flow-based generative models and their application to time series forecasting.  I may be missing important theoretical nuances or standard conventions in this subfield, My review should be weighted accordingly or potentially disregarded if it conflicts with expert opinions..

**Questions:**

Even if the theory is sound, what actionable insights does it provide? How should practitioners use these results to design better models, select hyperparameters, or understand limitations?

---

> ### Author Response · Authors · 2025-11-30
>
> Thank you for your positive and valuable comments. We address your questions in detail below.
>
> ### Weakness 1
> Our work is purely theoretical and does not include empirical studies to illustrate how the approximation bounds, generalization bounds, or convergence behavior manifest in practical settings. As stated in the global response, our primary aim is to establish a foundational theoretical framework for flow-based generative models in time series, rather than to benchmark or validate specific architectures. In the revision, we will clarify that the contribution is strictly theoretical and acknowledge the value of future experiments designed specifically to illustrate these theoretical predictions.
>
> ### Weakness 2
> You raise an important point about whether the guarantees we provide are tight or directly predictive of practical performance. Our intention is not to claim tight bounds or optimal constants, but rather to demonstrate that under our assumptions, the function class and sampling dynamics are sufficiently expressive and stable to achieve arbitrary accuracy. This type of result is standard in theoretical analysis of generative models, where the goal is to establish theoretical possibility, rather than to provide resource-optimal prescriptions.
>
> ### Question 1
> While our contribution is theoretical, it does offer several conceptual insights that can inform practical model design.
> 1. By framing the sampling process as an optimization problem, our analysis suggests that more advanced optimization methods such as Adam may improve the efficiency or stability of sampling, an avenue that has not been widely explored in flow-based time-series modeling.
> 2. The use of polynomial bases in our generalization analysis highlights the role of structured regularization: although the specific polynomial construction is used as a theoretical tool, the underlying insight is that designing or tuning regularization terms that promote smoothness or controlled complexity can help models generalize better in noisy time-series regimes.
>
> We appreciate your detailed reading and constructive suggestions.

---

### Official Review · Reviewer_M7J3 · 2025-10-30

**Soundness:** 2
**Presentation:** 2
**Contribution:** 2
**Rating:** 2
**Confidence:** 4

**Summary:**

The author proposes a novel theoretical framework to verify the approximation, generalization and efficiency of flow-matching method within the time series generation task.

**Strengths:**

1. The motivation is clear and easy to follow, less typos
2. The background and related work are enough
3. The  paper gives a novel perspective for time series generation task.

**Weaknesses:**

1. As stated in lines 78-80, the paper propose to ensure robustness against noise and distribution shifts, where are the proofs to verify the robustness of distribution shifts of time series analysis, can author give some experiments ?

2. As stated that "the fitting of the flow-based generative models is confirmed to converge to arbitrary error under the universal ap-
proximation of Diffusion Transformer (DiT)", can author do the DiT structure-based flow model to verify this claim, such as make the comparison with Diffusion-TS, which is described in line 220.

3. As stated that "Orthogonal polynomial bases" is more stronger approximating ability,  can author make some experiments to verify the effectiveness?

**Questions:**

please refer to the weaknesses.

---

> ### Author Response · Authors · 2025-11-30
>
> Thanks for your insight review and recognizing the novelty of our idea. Our responses to your questions are provided below.
>
> ### Weakness 1
> Our theoretical results are derived under the data model and noisy sampling assumptions stated in Section 3, and the robustness we prove is with respect to noise and variability within any distribution that satisfies these assumptions, not arbitrary out-of-distribution or adversarial distribution shifts. In particular, our guarantees apply to any time-series data distribution fitting the Sobolev-RKHS and noise conditions of Definition 3.1, and thus provide a theoretical explanation for why diffusion and flow based time series models, which have already shown strong empirical performance in prior work, can succeed under such settings.
>
> ### Weakness 2
> As we mention in the global response, we acknowledge that our work is purely theoretical and does not present experiments. And showing empirical correspondence to theory is an interesting direction, but it is outside the intended scope of this submission. We will add a limitation section to our work to clearly communicate that the contribution is conceptual and theoretical, not empirical.
>
> ### Weakness 3
> In this work, the orthogonal polynomial bases are introduced primarily as a theoretical tool to obtain clean approximation and generalization guarantees. As mentioned in the global response, given the over-parameterization and universal approximation properties of DiT models, there typically exist many parameter configurations that can approximate the target functions well, and in practice larger models with sufficient regularization and data tend to further improve performance rather than harm it.
>
> We appreciate your detailed reading and constructive suggestions.

---

### Official Review · Reviewer_9KCz · 2025-11-02

**Soundness:** 3
**Presentation:** 2
**Contribution:** 3
**Rating:** 6
**Confidence:** 1

**Summary:**

I am very sorry, but I don't have the background to be able to understand this paper. I've tried to at least connect the main theorems, but even that it is very hard. According to the abstract the paper provides a formal framework that allow them to prove guarantees related to approximation, generalization and efficiency. If correct, it would be a valuable contribution.

**Strengths:**

Providing formal guarantees for complex machine learning is an important problem.

This is a very formal paper, under the assumption that it is correct, this is a strength.

**Weaknesses:**

The paper is not accessible for people without deep mathematical understanding of the topic (this could of course be perfectly fine).

It is hard to get an intuitive sense for the formal framework and theorems.

There are many transformations between the many lemmas and theorems which makes it hard to even trace the connection between them.

**Questions:**

The framework is focused on time series models, would it be possible to use the same framework for non-sequential models?

If I understand correctly, the method is based on diffusion models, what would it take to adapt it to transformers?

---

> ### Author Response · Authors · 2025-11-30
>
> Thank you for your positive and valuable comments. We address your questions in detail below.
>
> ### Weakness
> Thank you for your understanding, we fully understand your concern. This level of technical depth is largely inherent to the contribution we aim to make, our results rely on a sequence of structural assumptions and intermediate lemmas that build toward the approximation, generalization, and efficiency guarantees.
>
> ### Question 1
>
> Yes. For example, language data can be interpreted as a one-dimensional time series, and after embedding, each embedding coordinate becomes its own continuous trajectory. Under this representation, the model operates on a multivariate time series, and our theoretical framework applies dimension-wise in the same manner as in the standard setting. This perspective aligns with how diffusion-based language models treat token embeddings as continuous sequences. Therefore, while our analysis is developed for continuous, time-indexed data, it can extend to modalities that can be represented as collections of continuous sequences with comparable regularity properties.
>
> ### Question 2
> We would like to clarify that our approximation analysis already uses the Diffusion Transformer (DiT) architecture. The universal approximation theorem we rely on (as noted in Section 5) explicitly concerns transformer-based networks.
>
> Thus:
> No adaptation is needed—the flow-matching objective is already parameterized via a transformer-based diffusion model. The analysis directly incorporates transformer properties to establish approximation guarantees.

---

### Author Response · Authors · 2025-11-28

We sincerely thank all four reviewers for their careful reading and constructive feedback on our work. We appreciate reviewer 9KCz for recognizing the importance of bringing formal theoretical guarantees to flow-based generative modeling. We thank reviewer M7J3 for highlighting both the clarity of the motivation and the novelty of our theoretical perspective. We appreciate reviewer PYku for acknowledging the conceptual breadth of our unified theoretical framework. We thank reviewer 6YoR for appreciating the coherence and depth of our analysis.

Below, we want to address some concerns that were raised by multiple reviewers:
1. We want to restate that our goal is to establish a rigorous theoretical framework for generative models whose empirical effectiveness has already been demonstrated in prior work.
2. The orthogonal polynomial bases used in our analysis serve as a theoretical device to obtain clean approximation and generalization guarantees, not as a claim of empirical superiority.
3. Given the over-parameterization and universal approximation properties of DiT models, there exist many solutions capable of accurately modeling the target functions, and in practical settings, larger models with suitable regularization and data typically improve, rather than hinder, performance. Our focus in this paper is therefore on articulating the theoretical foundations underlying this behavior, rather than duplicating empirical evidence already well established. We have also added a limitation section to our work to discuss the concerns above.

---

### Meta-Review · Area_Chair_yQiX · 2025-12-30

**Summary:**

This work is a theoretical study of flow matching for time-series generation. Reviewers agree that this is a timely and important direction.

Reviewer opinions are mixed, but their evaluations are generally high-level and low confidence, with several reviewers explicitly noting limited expertise in the paper's area. A common criticism raised by the reviewers is a lack of empirical results, but the reviews generally lack substantive discussion of the paper's theoretical contributions.

Based on my own assessment, while the paper appears technically sound, the theoretical contributions are often incremental and not sufficiently well-motivated or contextualized relative to existing results. In particular, the paper does not clearly articulate the new insights provided by the theory or why these results meaningfully advance understanding in the area. For instance:
- As noted by several reviewers, the role of the polynomial regularization has an unclear connection to practice, and this aspect is not sufficiently discussed in the paper
- Several of the main theorems are direct applications of known results (Theorem 4.5, Theorem 4.9, Theorem 5.4)
- The main stated theorems generally lack a discussion of their significance or interpretation (Theorem 6.2 Theorem 7.7)
- Some theorems are stated without a sufficiently complete or formal proof (Theorem 6.2)

Given the lack of strong, expert support in the reviews and the limited evidence of significance beyond correctness, I do not feel there is a sufficiently compelling case for acceptance at this time. I believe the paper would benefit from further development and clearer positioning before being recommended for acceptance.

**Reviewer Concerns:**

- Reviewers universally are concerned about a lack of empirical results, but no results were provided in the rebuttal
- Reviewers comment on the relevance of the polynomial smoothing, but no significant discussion of the relevance of this technique to practice is discussed

**Reviewer Scores:**

Reviewers seem unlikely to have updated their scores given their overall lack of expertise in this area.

---

### Decision · Program_Chairs · 2026-01-26

Reject